# The Predictability of the Dental Practitioner in a Volatile Healthcare System: A 25-Year Study of Dental Care Policies in Romania (1999–2023)

**DOI:** 10.3390/healthcare13030249

**Published:** 2025-01-26

**Authors:** Ana Cernega, Dana Galieta Mincă, Florentina Ligia Furtunescu, Ciprian-Paul Radu, Simona Pârvu, Silviu-Mirel Pițuru

**Affiliations:** 1Department of Organization, Professional Legislation and Management of the Dental Office, Faculty of Dental Medicine, “Carol Davila” University of Medicine and Pharmacy, 17-23 Plevnei Street, 020021 Bucharest, Romania; silviu.pituru@umfcd.ro; 2Discipline of Public Health and Management, Faculty of Medicine, “Carol Davila” University of Medicine and Pharmacy, 050474 Bucharest, Romania; dana.minca@umfcd.ro (D.G.M.); florentina.furtunescu@umfcd.ro (F.L.F.); paul.radu@valinet.ro (C.-P.R.); 3National Institute of Public Health, Faculty of Medicine, “Carol Davila” University of Medicine and Pharmacy, 050474 Bucharest, Romania; simona.parvu@umfcd.ro

**Keywords:** doctor–patient relation, healthcare system, oral health policies, costs, quality, access to healthcare

## Abstract

**Background/Objectives:** This paper brings to light the results of extensive research on the dimension of dental care in Romania in terms of priorities and policies defined at the governmental level for a 25-year period (1999–2023). This research aims to present both individual and ecosystem perspectives on the influence of the way oral health strategies are shaped in Romania. **Methods:** The data collected are analyzed through the prism of the “iron triangle of healthcare”, focusing on the evolution of dental care from the perspective of the interrelationship of three important components: cost, quality, and access to healthcare services. **Results:** The paper provides insight and clarity into the context in which the doctor–patient relationship is constructed and developed, as well as a vision of how oral health policies should be designed to meet the individual needs of the patient, which will have an impact on the health and quality of life of the whole community. **Conclusions:** This study emphasizes the need to reconfigure oral health policies in Romania. Despite some input achievements like a considerable number of dentists, the systemic challenges that developed over these 25 years (such as insufficient funding, the reduced reimbursed procedures, and the limited dentists contracting with NHIH) have significantly contributed to a stagnation or decline of the oral health outcomes at the societal level.

## 1. Introduction

At the core of a well-functioning healthcare system is the effective management of all available resources to ensure the delivery of health services to beneficiaries, which includes *the management of human*, *financial*, *physical*, and *time resources*.

In a world characterized by volatility, uncertainty, complexity, and ambiguity [1], the financing of healthcare systems has become one of the greatest challenges. This challenge relates, in particular, to ensuring a balance between the economic efficiency of health service delivery and the ethical and legislative components that guarantee the legal rights of citizens, with a focus on the universal right to health protection [2].

The World Health Organization (WHO) describes health as, a state of complete physical, mental and social well-being and not merely the absence of disease or infirmity” [3]. This definition presents a dual meaning of health and the application perspectives of government policies, individual health components, and the ecosystemic health component of a group or community [4,5].

***The individual component*** refers to the citizen as a patient and their integration into the foundational cell of the system, represented by the doctor–patient relationship. This relationship is based on the patient’s rational need to improve his state of health; this need is translated into the responsibility of the doctor; who has the legal authority and the necessary competence; to focus on three complex actions within the medical act as follows:To ensure *prevention* through the fundamental mission of informing and educating the patient;To establish an accurate *diagnosis*;To provide *treatment*, including management of potential complications.

***The ecosystemic component*** refers to the protection of human health at the macro level, encompassing the totality of doctor–patient relationships as well as the bodies and institutions that shape the functioning, outcomes, and impact of these relationships. WHO provides clarity on this ecosystem component and the perspectives through which a health system can be viewed and analyzed as follows:▪The set of activities whose principal aim is to promote, restore, and/or maintain health;▪People, institutions, and resources are deployed together in accordance with established policies to improve the health of the population they serve while meeting people’s legitimate expectations and protecting them against the costs of illness through a variety of activities whose primary intention is to improve health [6].

Thus, this ecosystem component is the set of health measures, strategies, policies, and programs promoted at the governmental level to ensure the protection of the constitutional right to health [7].

In Romania, Law 95/2006 on health reform regulates the following areas of public health intervention: prevention, surveillance, and control of communicable and non-communicable diseases; health monitoring; health promotion and education; occupational health; environmental health; primary and secondary regulation in the field of public health; public health management; specific public health services; specific medical services and treatment of diseases with a major impact on public health [8]. These areas of intervention are integrated into the health programs and policies developed and implemented by the Ministry of Health (MoH), with the participation of the National Health Insurance House (NHIH), which organizes the social health insurance system throughout the country. By social health insurance, we mean the right of the insured to have access to a package of services, including a range of *dental services*, as part of the financing of curative treatment. The inclusion of dental services is important, correct, and effective in terms of the implications it has: epidemiological, economic, legal, and the correlation it has with systemic health, but also in terms of the connection it can have with other medical areas [9].

National legislation provides for the implementation of measures to preserve and improve the oral health of the population, but studies carried out in Romania at different time periods provide a different perspective. The first study identified in the literature was published in 1997 and revealed a high prevalence of untreated dental caries in the 18–24 age group. 72% of the subjects aged 25–44 years had gingival bleeding and calculus, and only 24% had visited a dentist in the previous 12 months [10]. Another study published in 2014 showed an increase in the prevalence of oral diseases, with the prevalence of dental caries reaching 75% [11]. In 2017, 91% of children showed mild inflammation and gingivitis prevalence [12]. According to another study published in 2019, the prevalence of dental caries in Romania is 96.3% in young people and 97.5% in the adult population [13].

These studies also provide insights into the factors that lead to such poor oral health, such as the impact of social variables such as economic, environmental, and lifestyle factors, low socioeconomic and educational levels associated with low access to dental services, and poor oral hygiene behaviors [13,14,15].

At the same time, another factor contributing to the deterioration of the oral health of the Romanian population is the way in which dental services are paid for. The vast majority of dentists work in the private sector, and far too few of them provide dental services under contract with the NHIH, which puts great financial pressure on patients [15,16,17]. This issue particularly impacts patients with low socio-economic status, who are more likely to experience dental disease [18,19] and are consequently more prone to seeking dental care only during emergencies [15,20]. Consequently, from the point of view of patients’ rights, this financial burden caused by the need to cover the cost of treatment out-of-pocket can be interpreted as a restriction on access to dental services [9,21].

Despite continued increases in public health spending [22] and the implementation of reforms and policies under the NHIH, dental services remain a low priority in terms of applied policies and budgetary allocations [23,24]. This is evident in the disparity between the number of dentists working in the public and private sectors, which is a significant barrier to equitable access to dental care. Therefore, this study aims to analyze the evolution of dental care in Romania over the last 25 years (1999–2023), focusing on government priorities and strategies and their impact on the predictability of dentists in a volatile healthcare system. The null hypothesis is that the lack of strategic prioritization of dental care does not have a significant impact on the status of dental care in society.

## 2. Materials and Methods

Considering the above-mentioned aspects, as well as the lack of a comprehensive analysis of the evolution of dental services and their financing by the NHIH from 1999-2023, the hypothesis underlying this research was designed. The manuscript presents the way in which the dimension of the financing of dental services has been configured for the period 1999–2023 (25 years) in the context of the predominantly private sector orientation of the professionals, considering the impact on the oral health of the population. The research was conducted between March and June 2024.

To achieve these objectives, a multi-faceted approach was employed, combining the analysis of legislative frameworks, public documents, and statistical data.

The following indicators have been used as a structure and guide for the research:▪The numerical evolution of dentists in Romania over a period of 25 years: The total number of dentists practicing in Romania over the 25-year period was analyzed. Data were obtained from National Institute of Statistics (NIS) reports. In order to highlight trends in professional orientation and their impact on public access to dental care, the distribution of dentists between the public and private sectors was evaluated.▪Numerical evolution of dentists in contractual relations with the NHIH: The focus of this indicator is on the number of dentists providing services under contract to the NHIH. Data were extracted from the annual reports of the NHIH. The aim of the analysis was to understand whether NHIH contracting has increased, decreased, or remained stagnant and to assess the accessibility of publicly funded dental services and potential inequalities between public and private provision.▪Annual evolution of public expenditure on dental care financed by the NHIH over a period of 25 years: This involved a detailed examination of the budgetary allocations made by the NHIH to dental services over the 25-year period. Public expenditure data were collected from NHIH financial reports. Comparisons were made with total health expenditure to understand the prioritization of dental care within the broader health system.▪Evolution of the package of dental services, type and number of services, type of beneficiaries, according to the framework contracts from 1999 to 2023: The data were collected from government decisions approving the framework contracts on the conditions for the provision of medical care within the social health insurance system, as well as from ministerial orders of the MoH approving the methodological norms for the application of these framework contracts during the same period.▪Annual evolution of the lump sum (fixed settlement ceiling budget) paid by NHIH to physicians for dental services: Data on the fixed budget ceiling for dental services were also collected from the above-mentioned government decisions and orders of the MoH regulating the provision of health services.

Through the analysis of the above-mentioned indicators, the research aimed to clarify the extent to which the strategic prioritization of dental services has an impact on the oral health of the population. The approach ensures that the results present both the health system perspective on the evolution of dental care (ecosystem component) and an understanding of the influence and impact that this evolution has had on the constituent cell and on each of the two actors (individual the doctor–patient relationship).

## 3. Results and Discussions

### 3.1. The Numerical Evolution of Dentists in Romania

In order to analyze the evolution of the number of dentists in Romania, the “Tempo Online” application of the NIS was used [25], which generated statistical data for the period 1999–2023. It is important to mention that these data reflect the registration status of dentists and does not specifically indicate their active work status or whether they are currently practicing within the country.

Regarding the number of dentists in Romania, we can observe a continuous increase in the number of dentists, from 7708 dentists in 1999 to 21,200 dentists in 2023. In percentage terms, their number has increased by 175% over this 25-year period (Figure 1).

#### 3.1.1. Numerical Evolution of Dentists According to Practice Ownership (Public vs. Private)

In 1999, the distribution of dentists according to the system in which they practiced was as follows: the vast majority practiced in the public system (5261 dentists), representing 68% of the total number of dentists. This discrepancy can be explained from the perspective of a characteristic reminiscent of the old centralized socialist health care system introduced during the communist period, characterized by a high degree of centralization, the prohibition of private activity, and the concentration of medical personnel in the public system. According to Figure 1 presented above, the situation changed in 2005, when the number of private dentists doubled compared to 1999, accounting for *54%* of the total number of dentists. The number of dentists in the public system was in a continuous decrease until 2019 (1711 dentists), then increased until 2022, reaching 4677 dentists.

Decentralization and the legal regulation of the possibility of working in the private system have contributed to an upward trend in the number of dentists choosing the private sector, from 2447 in 1999, representing *32%* of the total, to 17,429 in 2023, representing *82%* of the total. In percentage terms, this represents an increase of *612.25%* over 25 years (Figure 1).

It is important to note that the data on dentists working in the public and private sectors, as extracted from the TEMPO online database, reflect their primary sector of employment but do not account for dual practice, meaning that some dentists may practice in both sectors simultaneously.

#### 3.1.2. Numerical Evolution of the Number of Inhabitants per Dentist and of the Number of Dentists in Relation to the Population (10,000 Inhabitants)

The statistical indicators that highlight the extent of the burden on the health workforce are the number of inhabitants per dentist and the number of dentists in relation to the population. For example, the NSI presents data for the period 2000–2023, showing that the numerical concentration of inhabitants over this period decreased from 2701 inhabitants/dentist in 2000 to 899 inhabitants/dentist in 2023, a percentage decrease of *67%.*

The number of dentists in relation to the population increased from 3.7 dentists/10,000 inhabitants in 2000 to 11 dentists/10,000 inhabitants in 2022, an increase of *197.3%* in 24 years.

These changes are mainly due to the numerical increase in the number of professionals providing dental services, as can be seen in the subchapter analyzed above. This development is important in terms of the direct and certain benefits generated at the level of the patient as a beneficiary of healthcare services. Thus, a lower concentration of inhabitants/dentists means better access to dental services for patients and the possibility of allocating more time and using it more efficiently in the process of treating patients. At the same time, the waiting time from appointment to consultation and treatment is reduced, which directly contributes to the development of the quality component of treatment, guaranteeing both the maintenance and improvement of the oral health status of the population in general.

### 3.2. The Numerical Evolution of Dentists in Contractual Relations with the NHIH

It is of particular interest to analyze the evolution of the number of dentists in Romania who have contracted with NHIH. Under the terms of Law 95/2006 and the Framework Contract, dentists have the possibility, within the limits of the allocated funds and the therapeutic acts covered, to establish a contract with NHIH, which means the provision of dental services for the population covered by the reimbursement.

Analyzing the activity reports of the NHIH data on the number of contracts in the field of dentistry could only be identified for the period 2018–2022. It can be observed that although there were numerical changes, slight decreases or increases compared to previous years (an increase of 1.33% in 2019, a decrease of 4.98% in 2020, an increase of 2.03% in 2021, and an increase of 4.12% in 2022) regarding the number of dentists contracted with the NHIH over a 5-year period, it is not possible to conclude the existence of an upward or downward trend due to insufficient data for longer periods. Rather, there seems to be a stagnation in the contracting process and in the interest of private dentists in establishing such contractual relationships.

An analysis of the data presented above shows that in 2022, only 3231 dentists had a contract with NHIH, representing ***19.2%*** of dentists practicing in the private sector.

Although the approach underlying the policies and strategies applied at the level of the health system is based on the principle that health and the measures taken to maintain it are a fundamental right of the individual and not a privilege granted, it appears that there are several external factors that have led to a stagnation in the increase in the number of dentists contracted by the NHIH.

In the meantime, access to health care services in the private system represents an economic burden for the patient due to the small number of reimbursable service providers in the country. In the context of financial incapacity, the patient may not be able to pay, leading to the postponement of planned treatment, with a consequent negative impact on the individual’s health status.

### 3.3. The Budget of the National Health Insurance Fund (NHIF)

In order to provide a clearer perspective on the budgetary value of medical services, especially dental services, Table 1 shows the evolution of the Leu-Euro exchange rate. It should be noted that the National Bank of Romania has provided data since 2005, following the monetary reform of that year. This reform simplified the monetary system by replacing the old Leu with the new Leu at a ratio of 10,000 to 1.

In terms of budget allocations and expenditures, dental care is a subcategory integrated into the *budget chapter on medical materials and service provisions* [22]. According to data collected from an NHIH report that presents the annual evolution of the NHIF from 1999 to 2023 (Table 2), it can be observed that, in general, expenditures for the chapter on medical materials and service provisions have experienced a continuous upward trend, with a percentage increase of *2689%*, from *1548.2 million Lei* in 1999 to *43,179.3 million Lei* in 2023, but these are the absolute amounts not adjusted for inflation.

If we analyze the evolution of the expenditures specifically in the field of dentistry, we can observe an upward trend, from *39 million Lei* in 1999 to *284 million Lei* in 2023, representing a percentage increase of *628.2%* during this period. A year in which a drastic decrease in dental expenditure was recorded is 2013, with an expenditure of 13.2 million Lei, representing a decrease of *77.2%* compared to the previous year. According to NHIH reports, this amount was fully spent in the first quarter of 2013, and as a result, the health insurance system was unable to continue funding dental medical services until the end of the year. This situation occurred because the MoH intended to implement an oral health prevention program in 2013.

In addition to these aspects, it is important to analyze the evolution of *the percentage allocated to dental care* in relation to the total expenditures for the chapter on medical materials and service provisions. Although, intuitively, there should have been an upward trend in this regard, the actual situation is characterized by dynamism and volatility.

Although budget expenditures have increased in both cases, there is no observable correlation between the evolution of the percentage allocated to oral health and the evolution of the total budget, according to the data presented in Figure 2.

Thus, at the establishment of NHIH, expenditures for dental care represented *2.51%* in 1999, which continuously decreased until 2013, reaching only *0.06%*, and then slightly increased to a percentage index of *0.65%* in 2023. Overall, analyzing from the perspective of 25 years, although the budget for dental services expenditure has increased year by year, the percentage allocated to the dental care subchapter has experienced a decrease and stagnation, representing a *74.1%* reduction over a 25-year period.

### 3.4. The Evolution of the Monthly Ceiling Allocated to Dentists Contracted with NHIH

In accordance with the provisions of the Framework Contracts and their implementation guidelines, payment for the dental services provided by the dentists contracted by NHIH is made on the basis of a fee for service paid in local currency (Lei). The number of therapeutic procedures performed per month is determined and limited by the value of the total budget (monthly ceiling or lump sum) from the signed contract. Consequently, once the budget is used, the patients have to pay out of pocket for the dental services received.

The contractual ceiling is influenced by both the professional level of the contracted provider and the environment in which they practice. The main purpose of this measure is to promote the continuous learning and professional development of dentists and to encourage the equitable distribution of human resources in rural areas, thereby contributing to the provision of dental services to a population with limited access to health care. Thus, a primary care dentist will receive a 20% increase in the monthly allocated budget, while dentists who choose to provide dental care in rural areas will have their budget increased by 50%.

The documents we analyzed provide an overview of the evolution of the monthly ceiling per dentist for the period 2015–2023, as shown in Figure 3.

According to the data presented, it is possible to observe a stagnation rather than an evolution in the monthly ceiling allocated to a dentist for the provision of dental services. Thus, we can identify three periods: 2015–2017, with a monthly ceiling of 1600 Lei; 2018–2022, with a monthly ceiling of 2000 Lei; and 2023, when the ceiling was increased to 6000 Lei (a percentage increase of 200% compared to 2022).

These data highlight two major vulnerabilities in the healthcare system as it relates to the doctor–patient relationship: from the dentist’s perspective, there is a limitation on the number of reimbursable therapeutic procedures for beneficiaries, and from the patient’s perspective, it can be interpreted as a restriction of their access to dental services.

Although 2023 shows a tripling of the monthly ceiling, it must also be analyzed from the perspective of the dental tariffs applicable in 2023. Thus, this increase in the monthly ceiling does not necessarily represent an increase in the number of therapeutic procedures available to beneficiaries but rather an adjustment to the increasing tariffs for dental services.

### 3.5. The Analysis of the Evolution of the Dental Services Package

According to the provisions of Law 95/2006 on the reform of the health care system, insured persons benefit from a set of medical services, health care services, medicines, sanitary materials, medical devices, and other services, all of which are included in a basic package of services that is approved annually by government decision. The package of basic services to which insured persons are entitled is regulated by the framework contract drawn up by the NHIH, which also draws up the methodological rules for its application following negotiations with the Romanian College of Physicians, the Romanian College of Dentists, the Romanian College of Pharmacists, the Romanian Order of General Medical Assistants, Midwives and Nurses and the Romanian Order of Biochemists, Biologists and Chemists in the Romanian Health System.

In the field of dentistry, dental service providers working under contract with the NHIH are offered the opportunity to perform a fixed number of dental procedures. Therefore, in order to analyze the evolution of the basic package of services specific to the dental sector, the framework contracts and their methodological implementation guidelines for the period 1999–2023 were studied. The data collected are presented graphically in Figure 4, which shows a strong downward trend in the number of reimbursable therapeutic procedures for dental care. Two significant decreases can be observed: in 2012, the number of reimbursable procedures was reduced by 44, a decrease of *51.1%* compared to the previous year, and in 2014, when the number of reimbursable procedures was further reduced by 17, a decrease of *41.4%* compared to the previous year. From 2014, the number of procedures stopped decreasing and showed a slight increase from year to year, reaching 31 dental procedures in 2023. Overall, it can be seen that over a 25-year period, the number of dental procedures decreased three times, with a percentage decrease of *66.6%* in 2023 compared to 1999.

#### 3.5.1. Analysis of the Evolution of the Package of Dental Services in Relation to the Type of Beneficiaries

Law 95/2006 on health reform, in conjunction with Law 227/2015, provides a detailed list of those who qualify as beneficiaries of the services included in the basic package, distinguishing between those who contribute to the establishment of the NHIF through income received regardless of its category (salaries, pensions, independent activities, intellectual property rights, etc.) and those who benefit without the obligation to pay contributions (children up to 18 years old, apprentices or students, doctoral students engaged in teaching activities, young people up to 26 years old from the child protection system, persons with disabilities, etc.).

In the field of dentistry, the Framework Contracts and their implementation guidelines for the period 1999–2023 present in tabular form the distribution and coverage of the types of services included in the basic package according to the type of beneficiaries, which fall into three categories: children, adults, and beneficiaries of special laws.

It can be observed that the legislation in force during the period 1999–2004 granted the status of beneficiary of the basic package of dental services only to children and adults, with an extension in 2005 to beneficiaries of special laws. When analyzed individually, the numerical evolution of reimbursable dental procedures for children and beneficiaries of special laws follows a pattern similar to that discussed in the previous subchapter, with two points of decline observed in 2012 and 2014 as follows:▪Children category: gradual decrease in the number of dental procedures between 1999 and 2011, from 91 to 78; a percentage decrease of 48.7% in 2012 compared to 2011 and 43.5% in 2014 compared to 2013; a slight numerical increase from 22 reimbursable procedures in 2014 to 28 dental procedures in 2023; percentage decrease of 69.2% over 25 years.▪Beneficiaries of special laws category: percentage decrease of 60% in 2012 compared to 2011 and 57.5% in 2014 compared to 2013; a slight numerical increase from 14 reimbursable procedures in 2014 to 20 dental procedures in 2023; over a 19-year period, a percentage decrease of 73.6%.

Regarding the numerical evolution of reimbursable treatments for adults, three points of decrease can be identified as follows:▪Percentage decrease of 42.2% in 2004 compared to 2003 (from 45 to 26 therapeutic procedures);▪From 2003 to 2011, the number of therapeutic procedures increased slightly, followed by a percentage decrease of 34.3% in 2012 compared to 2011 (from 32 to 21 therapeutic procedures);▪The percentage decreased by 42.8% in 2014 compared to 2013 (from 21 to 12 therapeutic procedures).

In addition, it is interesting to analyze the strategy underlying the definition of the percentage of coverage for fees related to reimbursable procedures for adults during the period 2000–2002. This strategy is correlated with the attendance of the beneficiary adult at periodic prophylactic check-ups. These check-ups are conducted under the conditions and within the time limits established by the legal norms. Specifically, young people aged 16–20 are required to attend twice a year, while adults are required to attend once a year.

In this way, according to the methodological norms for the implementation of the framework contracts for the period 2000–2002, adults who attended a prophylactic check-up were entitled to 60% coverage by the NHIH, compared to 40% coverage for those who did not visit the dentist for a routine check-up. Such regulations can be considered genuine strategic incentives in the field of oral health promotion, aimed at promoting the idea of prevention, focusing on and shaping social thinking and mentality by guaranteeing financial protection measures that indirectly encourage healthy habits of regular visits to the dentist.

Similar preventive strategies are used in Germany, where patients’ co-payments are reduced if they undergo regular preventive check-ups documented in a ’bonus booklet’. Annual dental check-ups for five (or ten) years prior to treatment increase the amount reimbursed by the social insurance fund from 60% to 70/75% [26].

#### 3.5.2. Percentage Evolution of Tariffs Applied to Reimbursable Dental Procedures by NHIH

An important aspect in the analysis of the packages of dental procedures reimbursed by the NHIH, together with their numerical and percentage evolution, is the evolution of the tariffs applicable to these services. As we have seen above, the number of procedures has changed considerably and is gradually decreasing. The total package of dental services has changed through the elimination of therapeutic acts, the addition of new therapeutic acts, and the elimination of therapeutic acts presented individually by merging them with existing ones. In order to analyze the evolution of the tariffs reimbursed by the NHIH over this 25-year period, seven dental procedures were selected that remained in the basic package from 1999 to 2023. The tariffs applied for each year were centralized, and the percentage increase in these tariffs was calculated compared to the previous year. Considering the fact that on 1 July 2005, the currency reform was implemented, which replaced the old Leu (ROL) with the new Leu (RON), according to which 10,000 ROL = 1 RON, the tariffs specified in the framework contracts from 1999 to 2005 were specified in old Lei. In order to calculate the percentage increase, the tariffs were converted into RON.

In order to determine the percentage increase in applied tariffs at the annual level, the average percentage increase was calculated for the seven therapeutic procedures analyzed. The data obtained allow us to analyze the evolution of the percentage increase in applied tariffs in relation to one of the indicators that define the level of tariffs for goods and services in a country over a given period of time **inflation**, as shown in Figure 5 and Table 3.

In the context of the provision of health services, it is important to distinguish between the notion of tariffs and the costs that define the scope of these tariffs. Thus, if the costs represent the total individualized expenditure attributed to the provision of services (human resources, medicines, rent, equipment, maintenance), the tariff must be related to the reality of the costs [1]. Given the financial volatility of the world in which we live and work, a lack of cost awareness contributes to incorrect pricing.

Therefore, considering that inflation is the phenomenon characterized by the increase in the prices of goods and services or that the provision of dental services directly depends on a large number of services and products that ensure and guarantee the proper execution of the treatment plan, it is considered that the tariffs applicable in this area should aim to adapt to the fluctuations of the inflation index.

However, the data collected shows us disproportionate changes in the growth of dental tariffs in relation to the inflation index. We can see that there is a general stagnation in the evolution of prices, but we can also identify five moments characterized by anomalies in which the tariff increased at an accelerated rate: 2002, average percentage increase in tariffs of 43% compared to 2001; 2005, average percentage increase in tariffs of 20% compared to 2004; 2012, average percentage increase in tariffs of 129% compared to 2011; 2014, average percentage increase in tariffs of 62% compared to 2013; 2023, average percentage increase in tariffs of 60% compared to 2022.

Given that these price increases are not the result of an adjustment to changes in associated costs due to inflation, other factors and other related elements underlie the decision to increase them. Thus, correlating the data analyzed above, we can see that this increase in tariffs is, in fact, a consequence of the reduction in the number of reimbursable dental procedures: in 2012, the number of therapeutic procedures was reduced from 86 to 42; in 2014, the number of therapeutic procedures was reduced from 41 to 24.

### 3.6. Dentistry in Terms of Access to Services, Cost and Quality

The premise of this research was to build a comprehensive picture of the evolution of health services and their financing by the NHIH in the dental sector. This evolution has been characterized by a large number of systemic and legislative challenges resulting from the different historical moments through which the country has passed exit and adaptation from the communist system, the decentralization of the health system, the legislative adaptation with the integration into the European construction, the need to achieve correct management of the available resources, etc.

Based on the data collected, we propose an analysis of these data through the prism of the concept of “The Iron Triangle of Healthcare”, first defined by Dr. William L. Kissick in 1994 in the book Medicine’s Dilemmas: Infinite Needs Versus Finite Resources [27]. This approach provides a view of the complexity of the health dimension from the perspective of three essential buildings ensuring access to health services, ensuring high quality and at a cost defined by available resources, as shown in Figure 6.

The importance of these elements stems from the need to be aware of the influence that a change in one element can have on the others, either by decreasing or increasing them. Thus, in an unmanageable equilibrium, this triangle is an equilateral triangle that transforms into a scalene triangle that adapts according to the priority of the element considered. In his paper, Dr. Kissick argues that cost containment becomes a real challenge in the context of strategic priorities based on improving the quality of services provided. He also argues that increasing citizens’ access to health care would imply an increase in quality and an increase in costs [27]. In this context, through this research, we identify how the three specific elements of the iron triangle of health (*access*, *quality*, and *cost*) have been configured and influenced at the level of dental care in Romania during the 25 years under analysis as follows:▪The number of dentists has increased considerably, which can be understood and interpreted as facilitating patients’ *access* to health services while at the same time contributing to improving the *quality* component of the therapeutic act.▪However, the trend of dentists contracting with NHIH is stagnating (19% in 2022), which means that dentists have little interest in providing reimbursable services and are directing a large proportion of patients to private providers, implying direct out-of-pocket payments. The financial pressure on patients who are unable to pay in the private system leads to delays in treatment and possible deterioration in their health status, which, in terms of system-level policies, may translate into a possible reduction in their *access* to health services.▪The *budget* allocated to dental care has experienced an evolutionary increase, but we note an imbalance in terms of the evolution of the percentage allocated to dentistry in relation to the total budget allocated to health, which has experienced a continuous decrease, reaching a percentage share of only 0.65% in 2023. This dimension of the identified health costs reveals a lack of interest in prioritizing dentistry alongside other health areas.▪Although *tariffs* have increased, this increase is not the result of an adjustment to the effect of inflation on the cost of the therapeutic act but rather the result of a decrease in the number of therapeutic acts that can be reimbursed (↓66%). This may mean that beneficiaries have limited *access* to the dental services they need, which may also affect the *quality* dimension of medical care.▪In addition, the monthly ceiling allocated is low and insufficient to ensure the provision of services requested by patients, which may explain the low number of dentists contracting with NHIH.

The research conducted in Romania directed our attention towards identifying indicators previously studied in other countries. A review of the literature and public reports revealed data encompassing several of these indicators, which are detailed below.

WHO provides comparative information on the functioning of health systems in different countries of the world. For example, in terms of the number of dentists, the density of dentists per 10,000 inhabitants in Denmark will have decreased sharply from 13.1 in 1990 to 7.2 in 2020, with a total of 4166 dentists [28]. Dental care is free for people under the age of 22 and for vulnerable groups. Adults pay 35–60% of the cost. Treatment is provided by private practices with regional subsidies [29]. In 2022, spending on dental services accounted for 5% of Denmark’s total current health expenditure [30].

In France, the public health insurance system supports dental treatment through three coverage models: full coverage with a 70% reimbursement rate, partial coverage involving fixed fees with potential additional costs, and treatments that are not covered at all. Additionally, the national prevention program, “M’T dents”, offers free preventive check-ups and treatment for children and adolescents at specific ages. Recently, the program has been expanded to include pregnant women, providing services from the fourth month of pregnancy until twelve days postpartum [31]. In France, the dentist-to-population ratio has remained relatively stable over the years, shifting only slightly from 7.1 per 10,000 inhabitants in 1990 to 7 in 2021 [28]. Meanwhile, spending on dental services represented 4% of the country’s total current health expenditure in 2022 [30].

Norway’s dental care system is predominantly private, with 70% of dentists working in the private sector and most adults covering treatment costs out of pocket. However, the public dental system (PDS) offers free services for children up to the age of 19 and specific vulnerable groups, such as individuals with disabilities, the elderly, and refugees. Young people aged 19 to 20 benefit from reduced fees, while non-priority adults pay costs determined locally [32]. The dentist-to-population ratio saw a notable improvement, rising from 8.2 per 10,000 inhabitants in 1990 to 9.3 in 2021, reaching a total of 5020 dentists [28]. Furthermore, spending on oral health services accounted for 5% of the country’s health expenditure in 2022 [30].

The dental care system in Sweden has evolved significantly in recent decades, moving from a generously subsidized model to one that is more in line with economic realities while maintaining high-cost coverage and additional support for vulnerable groups [32]. Dental care is provided free of charge up to the age of 23. After that age, adults pay through a mixed system of government subsidies and personal payments [33]. In 2008, a new support scheme, ATB, was introduced, which offered adults a modest annual contribution towards basic care but focused resources on protection against high costs [32]. Patients pay the first €308 of treatment in full, then receive reimbursements of 50% up to €1584 and 85% for higher costs [24]. Vulnerable groups, such as individuals with disabilities or those in institutional care, benefit from free treatments, while county councils organize screenings and training sessions for staff in care centers [32]. Regarding the percentage allocated to oral health, it accounted for 5% of total healthcare expenditures in 2022 [30]. In 2021, the number of dentists reached 7973, marking a significant increase in their ratio per 10,000 inhabitants, from 14.3 in 1990 to 17.7 in 2021 [28].

Notably, according to the Organization for Economic Cooperation and Development (OECD), Estonia and Lithuania rank among the countries with the highest expenditure on dental care, dedicating 10% of their total health budgets to this sector [30].

Considering the identified data, future research could benefit from analyzing and understanding how the health systems of these countries are structured through the lens of the Iron Triangle of Healthcare, focusing on the interplay between access, cost, and quality to the approach used in our study.

In terms of limitations of our study, there is a lack of detailed data on the number of dentists under contract to the NHIH and their ratio to the number of patients in urban and rural areas. The study is also limited by the lack of information on doctors working in both the public and private sectors. The database that was used only provides information on the registration status of the dentists and does not provide any specific information on their active working status.

## 4. Conclusions

Aligned with the aim of this research, the results indicate that oral health policies and strategies in Romania require significant reconfiguration. While the country has a relatively large number of dentists, the funding strategy for this sector must be prioritized, particularly by increasing its share of the total health budget.

The findings reveal that, over the past 25 years, the strategic approach to dentistry has been characterized by limited funding, a decreasing number of reimbursable therapeutic procedures, and a low number of dentists engaging in NHIH contracts. These factors have directly contributed to a decline in the oral health status of the Romanian population.

The research demonstrates the interdependence of the elements of the health iron triangle and places the financing component of the dental system, managed by the NHIH, in the middle of the intersection of the three. We note the importance of designing the dental financing dimension, a component directly linked to the cost of health care so that it does not have a major impact on the other two elements and access to services.

Thus, in view of the data collected and analyzed above, we note the importance of defining and designing health policies and strategies (1) starting from the actors of the constituent cell of the health system doctor (2) and the patient (3), a relationship based on the complexity of the provision of dental services (4), as shown in Figure 7. Therefore, to improve the oral health status of the population and to ensure the highest possible compliance and participation of ***dentists*** in the provision of services in the public sector, the following proposals need to be implemented:▪Increasing the percentage allocated to dental services (5);▪Linking the tariffs to the reality of the costs associated with dental care (6);▪Increasing the number of reimbursed therapeutic procedures (7).

These proposals would bring a number of benefits from the ***patient’s point of view***, as follows:▪Increased access to dental services in the public system (8);▪Reducing the financial pressure of out-of-pocket payments in the private system (9);▪Which will help to increase patient confidence in the success of treatment (10).

Although we recognize that the scarcity of resources is a specific characteristic of costs and budgets, the implementation of these proposals will contribute to satisfying the patient’s rational need to maintain and improve oral and general health (11), to protecting the beneficiary from the financial risks associated with the disease and, consequently, to increasing the professional security of the dentist (12). This adaptation of policies and strategies in the field of oral health will initiate the shaping of the ecosystemic component of the health dimension, starting from ensuring the functionality of the individual component of the health system doctor–patient relationship (13). Ultimately, these changes would lead to the design of a balanced construction of the iron triangle of health at the level of the entire Romanian dental system.

## Figures and Tables

**Figure 1 healthcare-13-00249-f001:**
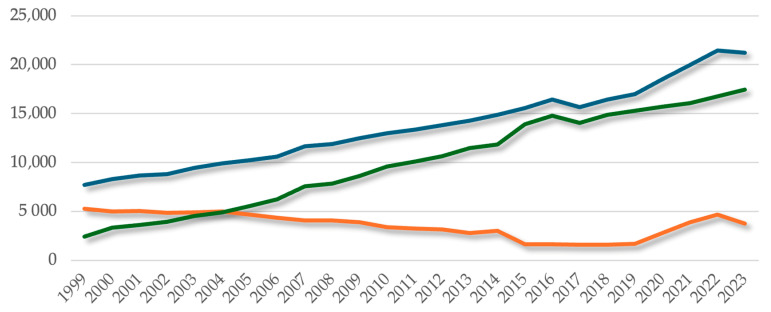
Numerical evolution of dentists of the period during 1999–2023.

**Figure 2 healthcare-13-00249-f002:**
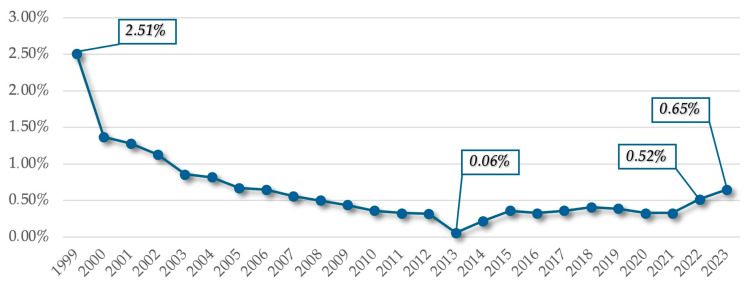
Percentage (%) allocated to dental care for the period during 1999–2023.

**Figure 3 healthcare-13-00249-f003:**
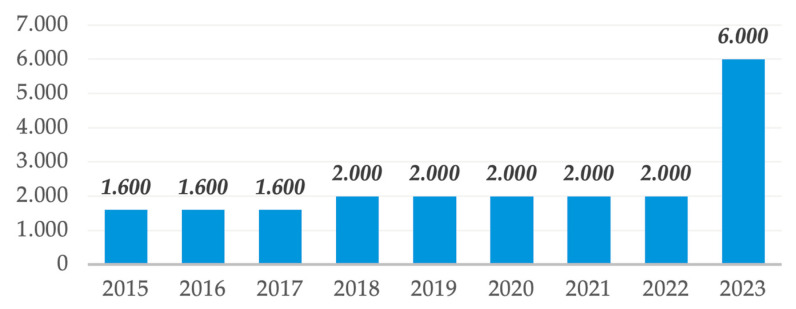
Monthly ceiling per dentist (Lei) during the period 2015–2023.

**Figure 4 healthcare-13-00249-f004:**
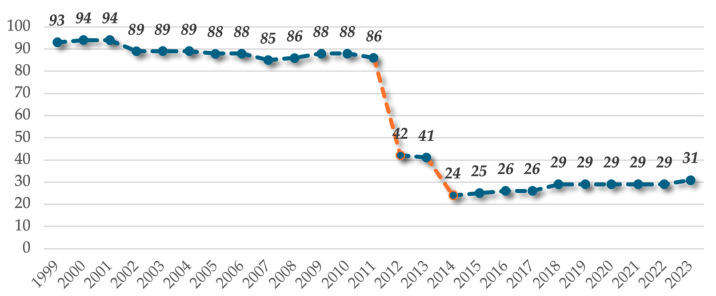
Graphical representation of the evolution of the number of therapeutic dental care procedures during 1999–2023.

**Figure 5 healthcare-13-00249-f005:**
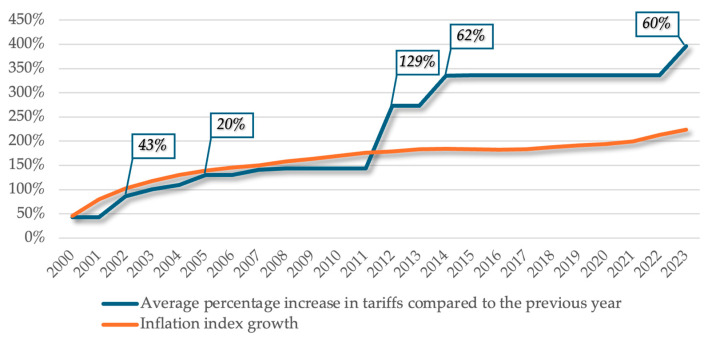
Graphical representation of the evolution of the average percentage increase in tariffs for dental treatment during 2000–2023.

**Figure 6 healthcare-13-00249-f006:**
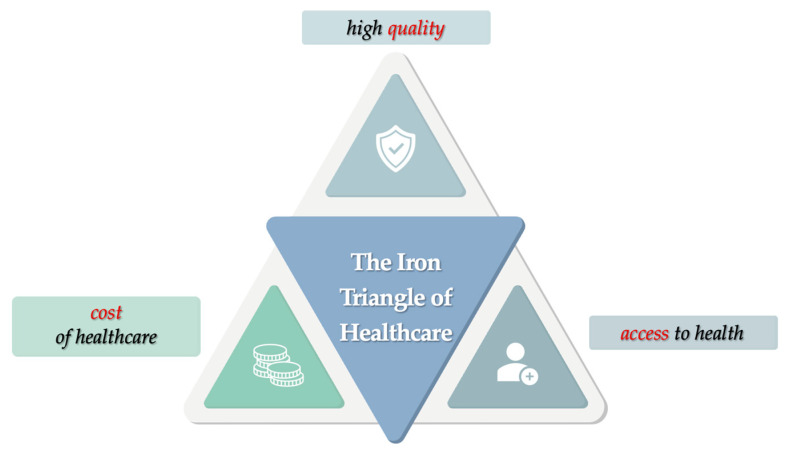
Graphical representation of the iron triangle of healthcare.

**Figure 7 healthcare-13-00249-f007:**
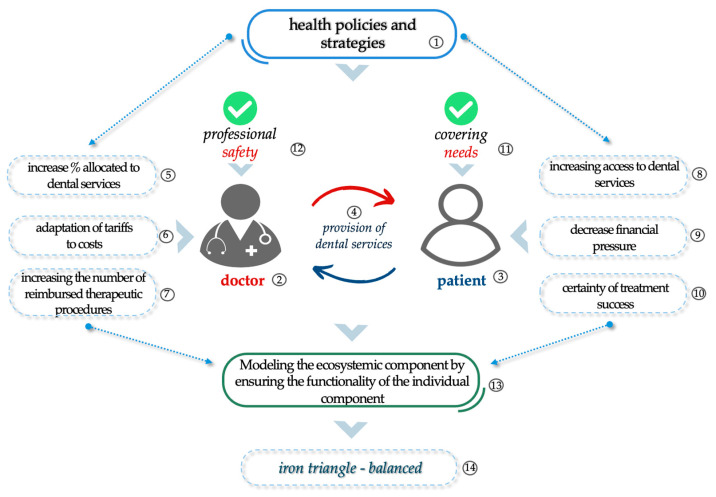
Strategies to balance the iron triangle of health in dentistry in Romania.

**Table 1 healthcare-13-00249-t001:** The evolution of the Leu-Euro exchange rate during 2005–2023.

*Year*	*2005*	*2006*	*2007*	*2008*	*2009*	*2010*	*2011*	*2012*	*2013*	*2014*	*2015*	*2016*	*2017*	*2018*	*2019*	*2020*	*2021*	*2022*	*2023*
Leu-Euro exchange rate	3.6234	3.5245	3.3372	3.6827	4.2373	4.2099	4.2379	4.4560	4.4190	4.4446	4.4450	4.4908	4.5681	4.6535	4.7452	4.8371	4.9204	4.9315	4.9465

**Table 3 healthcare-13-00249-t003:** Evolution of the average percentage increase in tariffs for dental treatment during 2000–2023.

*Year*	*2000*	*2001*	*2002*	*2003*	*2004*	*2005*	*2006*	*2007*	*2008*	*2009*	*2010*	*2011*	*2012*	*2013*	*2014*	*2015*	*2016*	*2017*	*2018*	*2019*	*2020*	*2021*	*2022*	*2023*
Average percentage increase in tariffs compared to the previous year	43%	0%	43%	15%	9%	20%	0%	11%	3%	0%	0%	0%	129%	0%	62%	1%	0%	0%	0%	0%	0%	0%	0%	60%
Inflation index growth	45.80%	34.50%	22.50%	15.30%	11.90%	9%	6.60%	4.80%	7.90%	5.60%	6.10%	5.80%	3,30%	4%	1.10%	−0.60%	−1.50%	1.30%	4.60%	3.80%	2.60%	5.10%	13.80%	10.40%

**Table 2 healthcare-13-00249-t002:** The Budget of the NHIF during 1999–2023.

*Year*	*1999*	*2000*	*2001*	*2002*	*2003*	*2004*	*2005*	*2006*	*2007*	*2008*	*2009*	*2010*	*2011*	*2012*	*2013*	*2014*	*2015*	*2016*	*2017*	*2018*	*2019*	*2020*	*2021*	*2022*	*2023*
Medical Materials and Service Provisions (million Lei)	1548.2	2481.7	3662	4750.3	6063.5	6894.8	9037.9	9521.2	12015.4	15628.6	14150.6	16104.7	16497.8	17999.9	21583	21184.2	21789.4	23930.1	23345.7	24950.5	29169.3	30261.2	33723.9	36860.9	43179.3
Budget Allocated to Dental Care (million Lei)	39	34	46.8	53.7	52.3	56.5	60.7	61.9	67.4	78.8	62.7	58.7	55.6	57.9	13.2	47.7	80	80.8	85.3	104.3	114.6	100.7	111.8	194.2	284

## Data Availability

The original contributions presented in this study are included in the article. Further inquiries can be directed to the corresponding author.

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
