# Peer review of "The Predictability of the Dental Practitioner in a Volatile Healthcare System: A 25-Year Study of Dental Care Policies in Romania (1999–2023)"

_healthcare, 2025, doi:10.3390/healthcare13030249_

Round 1

Reviewer 1 Report

Comments and Suggestions for Authors

I congratulate the authors for their very successful work. Unfortunately, there are not many scientific studies in the field of health that address the sociological and systemic aspects of the subject. For this reason, I believe that the relevant manuscript and study are important. However, some revisions should still be made to the manuscript.

1- The information about the country and time period where the research was conducted must be stated in the title.

2- The last paragraph of the Introduction section is one of the most important paragraphs for the reader. The relevant paragraph is difficult to understand and consists of only two long sentences. The paragraph should be organized in a more concise and understandable manner, and the purpose of the study and the null hypothesis should be stated clearly and explicitly.

3- The Material and Method section is prepared quite superficially, in direct contrast to the Introduction section. Remember that the most important section of a study is Methods. Therefore, please elaborate on the relevant section. I would especially like you to explain your indicators better and explain how and according to what you evaluated these indicators.

4- The Results section is extremely detailed, explanatory, comprehensive, and sufficient; congratulations.

5- The Discussion section must be developed. Comparisons should be made with dental health systems in different countries. Also, dental and medical system comparisons should be made. Also, if there are any, comparisons should be made with similar studies done in Romania. The Discussion section is insufficient in its current form.

Best wishes,

Reviewer 2 Report

Comments and Suggestions for Authors

Dear Authors!

Please let me congratulate for the nice work, however, I need to raise a few questions and issues.

Please add at least a short conclusion part to the abstract.

In Figure 1, Point 4, the number 4 is not visible, please correct the image.

Please also add the pints 1-5. To the figure caption. It is weird, that you can only find the meaning of this numbers in the above text.

Please explain, if figure 1 and its content is the authors own work and suggestion or it is a referred fact. It is is a referred fact, it shall be containing the reference. It it’s the authors own suggestion, it shall be in the results or conclusion part, otherwise how do we get this figure?

In line 99, you state: “ too few of them provide dental services under contract with the NHIH”. It would valuable, to have at least some data about the NHIH dentists-to patient ratio. Is there any data available?

In line 101-102 I suggest, that there is a connection of suffering from the disease and visiting the dentists in emergency. It there a connection, or the connection is the low socio-economic status? Please clarify or rephrase the sentence: “This particularly affects patients of low socio-economic status, who are more likely to suffer from dental disease [18, 19], as they are more likely to visit the dentist in an emergency [15, 20].”

In point 3.1. The Numerical Evolution of Dentists in Romania: Please clarify how it is controlled, if these dentists work also within Romania, or they are only registered? Is this database give data on registration or active work status?

In point “Numerical Evolution of Dentists According to Practice Ownership” you give numbers regarding private and public sector. Is there any overlap possible? For example a dentist working in the public sector can be a co-employed in the private in the afternoon/evening or weekend? If so, as it is in most EU countries, how did you calculate and count this part? Please clarify.

I do not see any data regarding the geographic properties of dentist patient so as patient to NHIH ration? This is an important factor, as most NHIH services are in most countries in the cities, and villages so as rural areas, where the economic need is even higher, do lack of the governmental services. Please add data or map. This is a crucial part f the whole understanding.

All the financial data is given in Lei. However, without data about GDP or continuous Lei-Euro rate, it is difficult to judge about any of these data. Please add relevant info.

Please rephrase the discussion part according to the new data added to the main text.

Reviewer 3 Report

Comments and Suggestions for Authors

This research focused on the dimension of the dental care in Romania in terms of priorities and policies defined at governmental level for a 25-year period (1999-2023).

It is an organized document with an unusual but interesting theme. This theme is rarely seen in the literature, and this justify the realization of an article. It is interesting to be able to know other countries healthcare system as well as proposals to improve the existing challenges in a volatile healthcare system.

Either way, I may have some comments in the various sections.

Introduction

I think the aim of this article should be pointed at the end of the introduction section, justifying what this document adds to the literature.

All text: when an acronym is introduced in the text please use only this In the sections next to the citation (e.g.National Health Insurance House (NHIH)- use this always, not the all name)

Material and methods- in this chapter you should define where you took the various information from (all the reports and so on). It must be published somewhere

Results:

all tables and figures should be indicated in the text, before (figures) or after (tables 1 and 2) they come in the strutur of the manuscript.

Comments of possible causes of specific results should be driven for the discussion chapter or include the results and discussion in the same subtitle.

Discusson- should include the limitations of this study and the ways to improve the study itself. I think the conclusion in a scheme that is presented in the conclusion should be included in this chapter

Conclusion- should be reformulated to respond to the objectives of the study.

Round 2

Reviewer 1 Report

Comments and Suggestions for Authors

I congratulate the authors for their successful work and revisions. The relevant manuscript is perfectly suitable for an article in this journal. 

Author Response

On behalf of our research group, I would like to thank you for your time and your comments. We appreciate your thoroughness and interest, and the fact that you have provided a lot of guidance in order to make a good article. We are honoured that you have taken the time and offered valuable advice to improve this article.

Reviewer 2 Report

Comments and Suggestions for Authors

Dear Authors!

Thanks for the corrections, tha paper has improved significantly.

I only noticed one problem: so far I know, Euro and Leu shall be written with capital letters. If I am right on this issue, please correct.

Author Response

On behalf of our research group, I would like to thank you for your time and your comments. We appreciate your thoroughness and interest, and the fact that you have provided a lot of guidance in order to make a good article. 

According to your suggestion we have corrected and changed Euro and Leu with capital letters.

We are honoured that you have taken the time and offered valuable advice to improve this article.